# A Noninvasive Risk Stratification Tool Build Using an Artificial Intelligence Approach for Colorectal Polyps Based on Annual Checkup Data

**DOI:** 10.3390/healthcare10010169

**Published:** 2022-01-17

**Authors:** Chieh Lee, Tsung-Hsing Lin, Chen-Ju Lin, Chang-Fu Kuo, Betty Chien-Jung Pai, Hao-Tsai Cheng, Cheng-Chou Lai, Tsung-Hsing Chen

**Affiliations:** 1Department of Information Management, National Sun Yat-sen University, Kaohsiung 804, Taiwan; chiehlee850427@gmail.com; 2Department of Emergency Medicine, Kuang Tien General Hospital, Taichung City 433, Taiwan; drsixmg@gmail.com; 3Department of Industrial Engineering & Management, College of Engineering, Yuan Ze University, Chung-Li City 320, Taiwan; chenju.lin@saturn.yzu.edu.tw; 4Division of Rheumatology, Allergy, and Immunology, Linkou Chang Gung Memorial Hospital, Chang Gung University College of Medicine, Taoyuan 333, Taiwan; zandis@gmail.com; 5Craniofacial Orthodontics, Craniofacial Research Center, Chang Gung Memorial Hospital, Chang Gung University, Taoyuan 333, Taiwan; pai0072@cgmh.org.tw; 6Division of Gastroenterology and Hepatology, Department of Internal Medicine, New Taipei Municipal TuCheng Hospital, New Taipei City 236, Taiwan; hautai@cloud.cgmh.org.tw; 7Department of Colon and Rectal Surgery, Linkou Medical Center, Chang Gung Memorial Hospital, Taoyuan 333, Taiwan; lai5556@cgmh.org.tw; 8Department of Gastroenterology and Hepatology, Linkou Chang Gung Memorial Hospital, Chang Gung University College of Medicine, Taoyuan 333, Taiwan

**Keywords:** *Helicobacter pylori* infection, colorectal polyp, teeth disease, precancerous lesions, non-invasive, risk stratifying tool, random forest

## Abstract

Colorectal cancer is the leading cause of cancer-related deaths worldwide, and early detection has proven to be an effective method for reducing mortality. The machine learning method can be implemented to build a noninvasive stratifying tool that helps identify patients with potential colorectal precancerous lesions (polyps). This study aimed to develop a noninvasive risk-stratified tool for colorectal polyps in asymptomatic, healthy participants. A total of 20,129 consecutive asymptomatic patients who underwent a health checkup between January 2005 and August 2007 were recruited. Positive relationships between noninvasive risk factors, such as age, *Helicobacter pylori* infection, hypertension, gallbladder polyps/stone, and BMI and colorectal polyps were observed (*p* < 0.0001), regardless of sex, whereas significant findings were noted in men with tooth disease (*p* = 0.0053). A risk stratification tool was developed, for colorectal polyps, that considers annual checkup results from noninvasive examinations. For the noninvasive stratified tool, the area under the receiver operating characteristic curve (AUC) of obese females (males) aged <50 years was 91% (83%). In elderly patients (>50 years old), the AUCs of the stratifying tools were >85%. Our results indicate that the risk stratification tool can be built by using random forest and serve as an efficient noninvasive tool to identify patients requiring colonoscopy.

## 1. Introduction

Colorectal cancer (CRC) is the most common cancer worldwide and a significant public health problem in developed countries [1,2]. Most CRCs arise from polyps considered to be precancerous lesions, particularly adenomatous polyps [3,4,5,6], even though most are asymptomatic. Removal of all precancerous lesions during endoscopy has been the most effective method for preventing cancer development [6,7,8]. Colonoscopy is the most effective method for the search and removal of colorectal polyps. However, colonoscopy is not only time consuming and costly but also has side effects. Previous studies have reported several adverse events of colonoscopy, including perforation (0.005–0.085%) and bleeding (0.0001–0.687%) [9]. These adverse events create health hazards for patients and financial burdens for healthcare centers.

Furthermore, the increasing demand for colonoscopy drastically increases the workload of gastroenterology [10]. The increasing workload might result in undesired results such as lower adenoma detection rates per colonoscopy [11] and longer waiting times for colonoscopy [12]. As shown in [12], the median waiting time for the screening colonoscopy is 210 days with the maximum waiting time equaling 631 days in Canada. Long waiting times increases the patient’s mental burden and the risk of precancerous polyps’ evolvement. Therefore, healthcare centers are actively searching for a risk stratification tool that identifies patients who require colonoscopy using noninvasive examination results.

Hence, risk factors of noninvasive examination data for colorectal polyps, such as gender, age, BMI, blood pressure, gallbladder (GB) polyp/stone, *Helicobacter pylori* infection, and tooth disease (periodontal disease, chronic gingivitis, and chronic periodontitis), were collected, and a machine learning method was implemented to build a risk stratification tool for patients with colorectal polyps. Risk factors were selected based on previous studies [13,14,15,16,17], which reported factors exhibiting some relationship with precancerous polyps [18]. Data from 20,129 consecutive asymptomatic individuals who underwent a health checkup were collected. To date, little is known about their association. Here, we hypothesized that noninvasive risk factors may be associated with colorectal precancerous lesions. Furthermore, we hypothesized that risk factors might vary from patients groups with different demographic characteristics such as gender, age, weights, etc.

After identifying noninvasive risk factors and patient grouping criteria, a noninvasive risk stratification tool was built in order to identify patients who need colonoscopy using a machine learning method. Previous studies have investigated the possibility of identifying patients at high risk for heart disease [19] and diabetes [20] using machine learning methods. More recently, artificial intelligence approaches such as machine learning methods have been used to build a risk stratification tool for different diseases [21]. Therefore, based on the identified risk factors, a machine learning method was further employed to show that the identified risk factors can serve as predictors of precancerous lesions.

To the best of our knowledge, this is the first investigation aimed at building a noninvasive stratification tool based on risk factors from annual checkup data. This study aimed to develop a simple, noninvasive, risk factor, and noninvasive risk stratification tool for these asymptomatic populations to determine colorectal precancerous lesions.

## 2. Materials and Methods

### 2.1. Study Participants

In this retrospective study, 20,129 consecutive asymptomatic patients who underwent a health checkup between January 2005 and August 2007 at Chang Gung Memorial Hospital (approval number: 201601348B0, approved 2016/01) were recruited. This study was approved by the Ethics Committee of the Institutional Review Board of Chang Gung Memorial Hospital and conducted according to the ethical principles of the Declaration of Helsinki, as reflected in the a priori approval by the institution’s human research committee. Written informed consent was obtained from all patients included in the study. Our health checkup program included physical examination, chest radiography, electrocardiography, complete blood tests, biochemical laboratory tests, urine analysis, abdominal ultrasonography, and colonoscopy. Exclusion criteria were patients who did not have colonoscopy during the course of the health checkup or had incomplete colonoscopy due to various reasons, such as poor bowel preparation or incomplete total colon inspection and BMI > 35 kg/m^2^. Height and body weight, used to calculate BMI, were measured by well-trained nurses. BMI ranges were underweight, under 18.5 kg/m^2^; normal weight, 18.5–25 kg/m^2^; overweight, 25–30 kg/m^2^; and obese, >30 kg/m^2^. In our institution, the C13 urea breath test was used to detect *Helicobacter pylori* infection [22].

### 2.2. Colonoscopy Procedure and Abdominal Ultrasonography

For bowel preparation, patients ingested 1.5–2 L of polyethylene glycol before the procedure. All procedures were performed by experienced gastroenterologists. Endoscopic findings were classified into two subgroups: polyp and polyp-free. GB polyps on ultrasonography showed fixed, hyperechoic material attached to the lumen of the GB, without an acoustic shadow [23].

### 2.3. Risk Stratification Tool Building

As described in Section 2.1, all items in the annual check-up data are collected for this research. Based on previous research [13,14,15,16,17], we selected risk factors from the following categories: (1) patient’s demographic characteristics including age, sex, weight, and height; (2) patient’s medical history including hypertension, diabetes, and *Helicobacter pylori* infection; (3) colonoscopy diagnosis results including colorectal polyps, ulcerative colitis, hemorrhoids, and intestinal hemorrhage, etc.; (4) abdominal ultrasonography diagnosis including GB polyps and GB stones; (5) blood sample diagnosis results including fasting blood glucose, total cholesterol, high and low-density lipoprotein (HDL and LDL), triglycerides, etc.; (6) dental diagnosis results including periodontitis, periodontal disease, chronic periodontitis, and chronic gingivitis. All diagnosis results are binary with respect to data with 1 = positive diagnosed and 0 = otherwise. BMI is calculated based on the weight of height of the patient. Furthermore, patients’ demographic data are dichotomized into binary or categorical data. Age is dichotomized as over (1)/under (0) 50 years old, and BMI is categorized as 0 (underweight (<18.5 kg/m^2^)), 1 (normal (18.5–25 kg/m^2^)), 2 (overweight (25–30 kg/m^2^)), and 3 (obese (>30 kg/m^2^)).

Our overall risk stratification tool building procedure is summarized in Figure 1 and *the Heuristic*.


*The Heuristic:*


Step 1:Collect data from annual health check-ups. All risk factors are indexed from *i* = 1…*N*, the value of the risk factor is *x_i_*, where there are *N* risk factors in total.Step 2:Pre-screen with a z-test for two sample proportions with a significance level equal to 0.05 is applied to select potential risk factors. Where the two sample proportions are calculated asFor all risk factor i,
*p_hi_* = the proportion of patients who has colorectal polyps for patients with risk factor *x_i_* = *h* − 1.
That is,
*p*_1*i*_ = the proportion of patients who has colorectal polyps for patients with risk factor *x_i_* = 0.*p*_2*i*_ = the proportion of patients who has colorectal polyps for patients with risk factor *x_i_* = 1.
Step 3:The null and alternative hypothesis is stated as below:Null Hypothesis: *p*_1*i*_ = *p*_2*i*_ = …. *p*_h*i*_
We record all risk factors which has a significantly different sample proportion between patients with and without colorectal polyps.
Step 4:Logistics regression is applied for each risk factor to calculate the discriminability for each risk factor. Based on the logistic regression, we identified the demographic risk factors which can segregate patients into different sub-groups for the machine learning process.Step 5:Machine learning is applied to each sub-group to construct the risk stratification tool.Step 6:We output the system of models which consisted of multiple random forest models.Step 7:Output our four-fold-cross validation.

### 2.4. Statistical Analyses

Statistical analyses, including receiver operating characteristic (ROC) curve, area under ROC (AUC), multinomial logistic regression analyses, and z-test for two-sample proportions, were conducted using SAS software (version 9.4; SAS Institute, Cary, NC, USA). We use the two-sample z-test for the pre-screen tool since it is simple and efficient. Researchers might consider another pre-screen method as well. Statistical significance was set at *p* < 0.05. Simple logistic regression was applied when the independent risk factor was binary (e.g., age), and multinomial logistic regression was applied when the independent risk factor was categorical (e.g., BMI). The AUC was reported for each logistic regression. Since underweight, overweight, and obesity groups were all considered abnormal, BMI was treated as categorical instead of ordinal data. Tooth disease was identified if the patient was diagnosed with periodontal disease, chronic periodontal disease, and/or chronic gingivitis. GB equaled a score of one if GB polyps and stones were observed on abdominal ultrasonography, whereas hypertension was based on the patient’s medical history and not the onsite measurement of blood pressure.

### 2.5. Machine Learning Algorithm

A machine learning algorithm, random forest, was adopted by using Python to build a risk stratification tool based on the risk factors identified from annual healthcare data. Discriminability was represented by AUCs. We used 75% of the data to build the model and 25% of the data to test the consistency of the model. The model building and testing process was repeated four times (four-fold validation method). Adulqader et al. [14] conducted a review on machine learning in healthcare. The authors point out the most popular classification method among all machine learning algorithms including support vector machine (SVM), random forest (RF), and Naïve Bayes. Previous studies [24,25,26] also use annual health check-up data to develop a risk stratification tool to serve as a screening tool for non-alcoholic fatty liver disease. Goldman et al. [25] use the decision-tree-based approach, and Fialoke et al. [26] used several other methods along with the decision-tree approach. We argue that since our risk factors are all binary data, a decision tree-based method such as RF is the most suitable method. Our machine learning algorithm is summarized as the following pseudo-code.

Machine Learning Algorithm (RF):

Step 1:Input all risk factors as vector X = <x1……xh> and the y = 1 if a patient is diagnosed with colorectal polyps, and zero otherwise. Moreover, input the demographic factors for aggregating patients into subgroups. Go to Step 2.Step 2:Segregate all patients into subgroups. Index subgroups as k = 1…N for N groups in total. Let k =1 and go to Step 3.Step 3:Input all risk factors X and y in the kth sub-group. Go to Step 4.Step 4:Input all data in with path_name = group k, with the following specification of random foreackage in python. We selected the four-fold validation, thus 75% of data will be randomly selected for modeling building and 25% will be reserved for validation. For each run, the random forest will repeat four times for validation. Output the model and go to Step 5.Branch criterion: gini indexNumber of estimators (number of decision trees): 1000Min_samples_leaf = 5Class weight: balancedValidation: Four-foldCalculate the following statistics:Specificity = True negative/(true negative + false positive)Sensitivity = True positive/(true positive + false negative)Area Under Curve (AUC)Step 5:Collected the outputted model and check if k = N, if not let k = k + 1 and go to Step 3, otherwise end the algorithm.

It is worth noting that all parameters are subjected to test and modified for different research topics. The parameters provided in the algorithm are the optimal parameters after our testing trials.

## 3. Results

### 3.1. Statistical Analysis

A total of 20,129 patients were enrolled, including 11,570 (57.5%) men and 8559 (42.5%) women, with a median age of 50 (range: 18–96) years, GB polyps/stones (3191, 15.85%), and tooth disease (15,346, 76.24%), as shown in Table 1. In this study, the risk factors of colorectal polyps were investigated. Each group was subdivided into two groups based on endoscopic findings: polyp and polyp-free. Logistic regression analysis was performed after adjusting for age, gender, BMI, GB polyp/stone, tooth disease, hypertension, and *Helicobacter pylori* infection to determine independent predictors of colorectal polyps. The prevalence of colorectal polyps was 27.08% (5450/20,129) and was associated with age, *Helicobacter pylori* infection, hypertension, and BMI (underweight and overweight) regardless of sex (*p* < 0.0001). Tooth disease only showed a significant difference in men (*p* = 0.0053), as shown in Table 2.

In Table 2, we find that the risk factors differ based on gender, age, and BMI. Therefore, all patients were divided into sub-groups based on gender, age, and BMI. For each group, risk factors for GB polyps, hypertension, tooth, disease, and *Helicobacter pylori* infection were input as independent variables to predict colorectal polyps. In Table 2 we presented the AUC of risk factors with *p*-values of the model and AUC from the logistics equations, where the *p*-values are less than 0.1 for at least male or female. Results of total cholesterol, high lipoprotein cholesterol, and triglycerides are excluded since their *p*-values are greater than 0.1. As we can observe from Table 2, the observed significances (*p*-values) for risk factors are different from male to female. Thus, we separate patients with their gender for the machine learning stage. While in Table 2 we did not examine the *p*-value for different BMI levels, previous literature suggests BMI might significantly relate to the evolvement of colorectal polyps. For example, [27] found that overweight and underweight statuses are significantly correlated with gut microbiota and metabolism. Jain et al. [28] found that obesity significantly impacts metabolism and is accessible with colorectal cancer and polyps. Hence, we also separate patients with their status of BMI.

Figure 2 and Figure 3 further demonstrate the significance and positive or negative impacts of each risk factor, respectively. In order to construct a risk stratification tool based on these risk factors, a random forest machine learning method was employed. In our study, age, *Helicobacter pylori* infection, and hypertension were all risk factors for colorectal polyps. A forest chart was also constructed to present estimated odds ratios for each risk factor, as shown in Figure 2 and Figure 3. While traditional statistical methods such as logistic regression have an AUC > 0.5, discriminability is not as high as healthcare centers may wish (0.5086–0.5900). Therefore, a machine learning method is required to build a model with higher discriminability. As shown in Figure 2 and Figure 3, abnormal body mass, age, and *Helicobacter pylori* are the most influential risk factors for colorectal polyps, regardless of the patient’s gender. We also found that hypertension was a significant risk factor for colorectal polyps in male patients. Moreover, the influence of different abnormal body masses was significantly different between gender and age groups. Thus, we further divided patients according to age, gender, and body mass to obtain 16 patient subgroups (2 × 2 × 4). Since risk factors differ according to age and sex, a risk stratification model was built for each group of patients. For each subgroup, a risk stratification tool was built via a machine learning method. Building a patient-characteristic-specific risk stratification model by using the machine learning method not only enhances the discriminability of the model but also identifies a set of more precise risk factors for each patient group. Healthcare centers can utilize these risk factors to precisely diagnose patients with colorectal polyps.

### 3.2. Noninvasive Diagnostics Tool with Random Forests

Based on our results in Section 3.1, we separate all patients into 16 groups via their age, gender, and BMI status. The random forest algorithm in Section 2.5 is applied to each group, and validation results are summarized in Table 3. The input risk factors include hypertension, chronic periodontitis, humanoids, *Helicobacter pylori* infection, GB stones and polyps, total cholesterol, high-density lipoprotein, triglycerides, and diabetes. However, not all risk factors are significant in the final model, and the performance of the stratification model varied extensively. In women < 50 years old with a BMI > 30 kg/m^2^, the random forest model’s discriminability (AUC = 91%) was high compared to that in other groups. The discriminability of detecting colorectal polyps is >80% for both women and men who are obese. The noninvasive detection tool has an AUC = 80% for underweight male who is >50 years old. In general, the noninvasive colorectal polyp detection tool has a higher AUC in patients with abnormal weight.

Furthermore, important risk factors identified by the random forests were examined. As shown in Table 3, in women aged >50 years and BMI > 18.5 kg/m^2^, the important risk factors are hypertension, diabetes, and GB stones. In contrast, in women <50 years of age and BMI >18.5 kg/m^2^, the important risk factors are GB stones and polyps. In men, for those >50 years of age and not underweight, the important risk factors are hypertension, diabetes, and high-density cholesterol. In men aged <50 years, the important risk factors are total cholesterol and high-density cholesterol. As observed, GB polyps and stones are important risk factors for predicting colorectal polyps in female patients.

## 4. Discussion

To the best of our knowledge, this is the first retrospective study to construct a noninvasive stratification tool for colorectal polyps based on an extensive set of risk factors identified by evaluating a possible association between colorectal polyps, GB polyps/stone, and tooth disease in healthy individuals. In this study, the participants were divided into two groups: polyp and polyp-free. Age, gender, BMI, GB polyps/stone, tooth disease (periodontal disease, chronic gingivitis, and chronic periodontitis), colorectal polyp, hypertension, and *Helicobacter pylori* infection; and triglyceride, high-density lipoprotein cholesterol, and total cholesterol were investigated. Upon disclosure, first, blood sugar status was not included since participants are required to offer their clinical data before checkup without the use of an invasive method such as “fingerstick” sampling to obtain the blood sugar level; second, the final pathological report of polyps was not illustrated because it was supposed that all polyps should be sampled for their nature to determine whether participants’ potentially have colorectal polyps, which are considered to be precancerous lesions [3,4,5,6].

An association was observed between the colorectal polyp group and age, *Helicobacter pylori* infection, hypertension, and BMI regardless of gender (*p* < 0.0001). Colorectal polyps (*p* = 0.0256) and BMI (overweight, *p* = 0.0111) were significantly different among female patients. Age, *Helicobacter pylori* infection, and hypertension were common risk factors for colorectal polyps.

Regarding age, many studies have reported the association between age and colorectal polyps [29,30], suggesting that CRC screening should be performed around the age of 50–60 years in the general population owing to >80% of CRCs being diagnosed over the age of 60 years, which is consistent with our results [31,32,33,34].

*Helicobacter pylori* infection is highly associated with hyperplastic polyps [34,35,36,37,38], fundic gland polyps [34], and colorectal polyps [16,39,40,41,42]. Physiological mechanisms are still unclear, although Meira et al. [34] reported that *Helicobacter pylori* infection is associated with chronic inflammation-induced DNA damage and increased levels of serum gastrin, and *Helicobacter pylori* CagA status may be the cause of colonic neoplasm formation [43,44,45,46].

Metabolic syndrome is characterized by the presence of at least three of the following five factors—abdominal obesity, elevated triglyceride levels, decreased high-density lipoprotein cholesterol levels, hypertension, and high fasting glucose levels [47]—and contributes to various diseases, including gastric neoplasm and colorectal neoplasm [48]. In our study, hypertension and BMI were significant across genders in our analysis, and as mentioned before, noninvasive methods are available for easily obtaining factor data from individuals before endoscopy. In our study, hypertension and BMI were both significantly associated with the presence of colorectal polyps.

As discussed in [27], BMI statuses, both overweight and underweight, can alter gut metabolism, and as [28] pointed out, the change in metabolism significantly relates to colorectal cancer and polyps. We hypothesis that BMI is a significant indicator for different colorectal health; therefore, the risk factor might change from one BMI status to another. The results of AUC prove that our hypothesis is correct. For some BMI status, it is easier to identify the patient with colorectal polyps and others are not. The risk factors also differ from one BMI status to another.

The bulk of data has validated dental problems as a risk factor for colon neoplasm development [15,49]. We surmise that periodontal disease may induce chronic inflammation, resulting in immune dysregulation, and alters gut microbiota, which could be one possible pathway responsible for colorectal carcinogenesis [50,51,52]. It was also found that GB polyps/stones are also related to colorectal polyps, consistent with recent studies [17,53]. This may be attributed to GB polyps/stones and colorectal polyps that share some risk factors, such as obesity and metabolic syndrome [54].

In our study, there is no doubt that all aforementioned risk factors are noninvasive indicators of colorectal polyp formation [48]. Our risk stratification tool, which is built based on identified risk factors with a machine learning method, exhibits high sensitivities (70–80%) compared with that in noninvasive tools developed by previous studies (60–70%) [55]. Other decision tree-based studies [25,26] build noninvasive stratification tools using annual check-up data for non-alcoholic fatty liver obtained in AUC ranges from 85 to 87%. Compared with previous studies, the proposed model outperformed in several subgroups, such as elder obsessive individuals.

The limitations of this study were as follows: (1) its retrospective nature; (2) it was conducted at a single institution with a Taiwanese population; (3) our sigmoidoscopy is conducted under anaesthetization. Thus, our dataset excluded patients with BMI > 35 due to the protocol code of the anesthesiologist. Future researchers can build an RF model for this subgroup or collect data of non-anesthetized sigmoidoscopy diagnostics.

## 5. Conclusions

In this research, we proposed a new approach for building a risk stratification tool for colorectal polyps. First, we identified a set of promising risk factors using traditional statistical analysis such as z-test and logistics regression. We find that risk factors significantly differ for different genders, ages, and BMI statuses. Then, we separate patients with key demographic characteristics, which we believe each subgroup has a different set of risk factors. Then, we implement random forest to build a machine learning model to stratify patients with and without colorectal polyps. Colonoscopy verification is warranted in those 50 years of age or older, with hypertension, and infected with *Helicobacter pylori*. However, colonoscopy verification is warranted in individuals with tooth diseases and GB polyps.

For obese females, GB polyps warrant further colonoscopy verification. For males over age 50 and not underweight, hypertension is a strong indicator of possible colorectal polyps. We also find that for either underweight or obese patients, the AUC is higher than other groups. That is, abnormal weight is a strong indicator of health status, and different health statuses should be modeled differently. This is verified by our design of grouping patients with different demographic characteristics before building a machine learning model.

Our risk stratification tool can help healthcare centers identify patients who need further colonoscopy. This tool provides two major benefits: first, it helps clinicians conduct colonoscopy and discover precancerous lesions earlier to prevent cancer; second, it reduces the time and financial burden of healthcare centers in conducting unnecessary colonoscopies.

## Figures and Tables

**Figure 1 healthcare-10-00169-f001:**
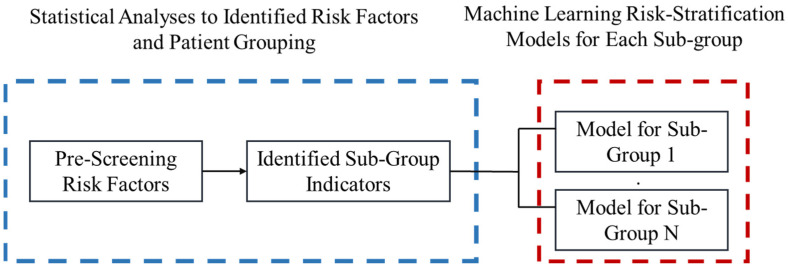
Diagram for proposed *Heuristic*.

**Figure 2 healthcare-10-00169-f002:**
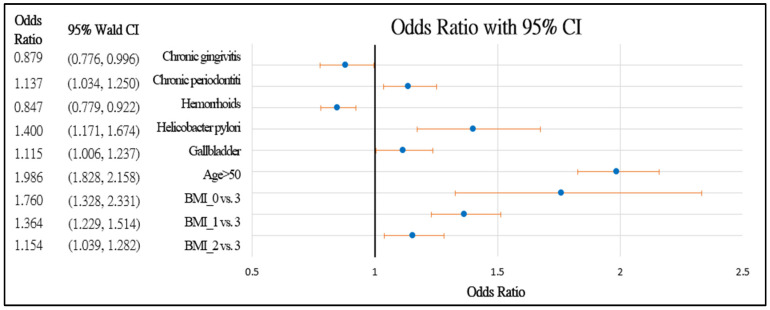
Forest chart of colorectal polyps’ risk factors in female patients. Underweight = 0, normal = 1, overweight = 2, and obesity = 3.

**Figure 3 healthcare-10-00169-f003:**
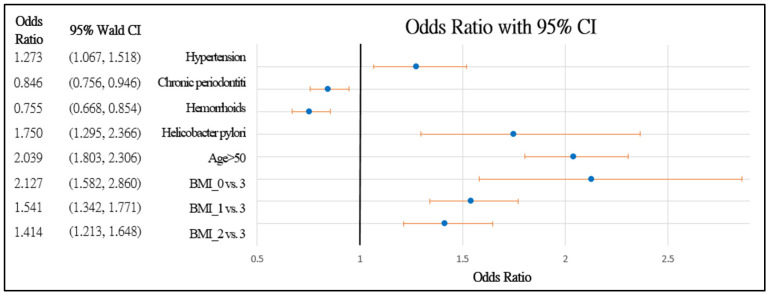
Forest chart of colorectal polyps’ risk factors in male patients. Underweight = 0, normal = 1, overweight = 2, and obesity = 3.

**Table 1 healthcare-10-00169-t001:** Participants’ clinical characteristics.

Total Number	*n*, %	20,129
Gender	Ratio of male to female (*n*/*n*)	11,570:8559
Polyp		
	Colorectal polyp (*n*, %)	5450, 27.08%
	Gallbladder polyps (*n*, %)	2188, 10.87%
	Gallbladder stone (*n*, %)	1106, 5.49%
Gallbladder problem		3191, 15.85%
Hypertension	(*n*, %)	1684, 8.37%
*Helicobacter pylori* infection	(*n*, %)	751, 3.73%
Tooth disease		15,346, 76.24%
	Periodontal disease (*n*, %)	8917, 44.30%
	Chronic gingivitis (*n*, %)	4168, 20.71%
	Chronic periodontitis (*n*, %)	11,655, 57.90%
BMI		
	Underweight (*n*, %)	805, 4%
	Normal (*n*, %)	9090, 45.16%
	Overweight (*n*, %)	6046, 30.04%
	Obesity (*n*, %)	4188, 20.81%
Age	Median (range)	50 (18–96) years
Total cholesterol		2818, 14%
HDL		2617, 13%
Triglycerides		3452, 17%

**Table 2 healthcare-10-00169-t002:** Multinomial logistic regression analysis of variables for colorectal polyps.

		Regardless of Gender		Male		Female	
Parameters		*p*-Value	AUC	*p*-Value	AUC	*p*-Value	AUC
**Age**	(>50 years = 1)	<0.0001	0.5847	<0.0001	0.5906	<0.0001	0.5900
** *Helicobacter pylori* **	(Yes = 1)	<0.0001	0.5113	<0.0001	0.5104	<0.0001	0.5092
**Hypertension**	(Yes = 1)	<0.0001	0.5142	0.0029	0.5084	<0.0001	0.5240
**Tooth disease**	Total	0.3734	0.503	0.0053	0.5118	0.1041	0.5086
**Gallbladder**	(Yes = 1)	<0.0001	0.514	0.002	0.5119	0.0185	0.5105
**BMI**							
	Underweight = 0	<0.0001	0.5604	0.0012	0.5389	<0.0001	0.5709
	Normal = 1	0.0055		0.1301		0.0341	
	Overweight = 2	<0.0001		0.0017		0.008	
	Obesity = 3						

**Table 3 healthcare-10-00169-t003:** Noninvasive stratifying tool (random forests model).

Gender	Age	BMI	Sensitivity	Specificity	AUC
Female	<50 years old	Normal	0.22	**0.74**	0.61
		Overweight	0.09	**0.83**	**0.76**
		Obese	0.14	**0.79**	**0.91**
		Underweight	0.55	0.50	0.66
	≥50 years old	Normal	0.35	0.66	0.68
		Overweight	0.27	**0.74**	0.68
		Obese	0.34	**0.74**	**0.85**
		Underweight	0.05	0.67	**0.79**
Male	<50 years old	Normal	0.38	0.68	0.63
		Overweight	0.39	0.59	0.68
		Obese	0.29	0.67	**0.83**
		Underweight	0.11	**0.72**	**0.75**
	≥50 years old	Normal	0.56	0.47	0.67
		Overweight	0.47	0.52	**0.70**
		Obese	0.43	0.57	**0.87**
		Underweight	0.28	0.65	**0.80**

## Data Availability

The data presented in this study are available upon request from the corresponding author. The data are not publicly available due to the protection of patients’ privacy and restriction from the Ethics Committee of the Institutional Review Board of Chang Gung Memorial Hospital.

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
