# Peer review of "A Noninvasive Risk Stratification Tool Build Using an Artificial Intelligence Approach for Colorectal Polyps Based on Annual Checkup Data"

_healthcare, 2022, doi:10.3390/healthcare10010169_

Round 1
Reviewer 1 Report
I would like to congratulate the authors for their fantastic work designing this project. It is a very interesting and promising tool.
I do have some concerns that need to be addressed:
- Introduction:
- colonoscopy is time-consuming, has several adverse events, and it is also worth to mention something about the waiting list and the load of work that a screening colonoscopy program means to the gastroenterologists.
- line 49: 'healthcare centres NEED TO identify patients who require colonoscopy using noninvasive examination' sounds too strong and there is no evidence for this until an appropriate tool is discovered and implemented. so please make a softer statement (i.e. it would be ideal, practical, etc).
- Authors don't mention if the included patients had previous polypectomies or not. is this an exclusion criteria? have you taken into account this when building the tool? Please clarifiy
2. METHODS
- seems that authors have excluded patients with BMI>35. please explain why
- It would be nice to have a list of all those risk factors taking into account for the analysis, before the identified ones.
3. Results
- the results of the 2 forest plot have not been explained within the results
- Table 2 should be better explained
4. DISCUSSION
- the discrepancies in results regarding BMI needs further explanation. why the authors think it has such an effect on the AUC? please discuss this factor
- a comparison between the authors' tool and previous ones developed deserved further discussion. what are the differences? what are the advantages of the authors' proposal?
5. Conclusion
- the authors don't mention the difference between male/female within the conclusion. do they recommend the colonoscopy verification in the same cases?
- And what about the BMI? does it need to be taken into account? please clarify
Author Response
Response Letter
Reviewer 1
2021/12/16
Dear Reviewer,
The authors would like to thank the reviewer for your effort of reviewing this manuscript. Your constructive comments help us to improve the quality of this paper. We would also like to wish the reviewer a happy Holiday Season! The reviewer’s comments are boldfaced, and our answer is in blue.
I would like to congratulate the authors for their fantastic work designing this project. It is a very interesting and promising tool.
A: Authors thanks reviewer for the comment. We appreciate your recognition! Thank you very much.
I do have some concerns that need to be addressed:
- Introduction:
- colonoscopy is time-consuming, has several adverse events, and it is also worth to mention something about the waiting list and the load of work that a screening colonoscopy program means to the gastroenterologists.
A: Authors thanks reviewer for the comment. To address how workload impact the patient and gastroenterologist we add the following paragraph in the introduction from line 50-58. Specifically, we add “Furthermore, the increasing demand for colonoscopy drastically increases the workload of gastroenterology [10]. The increasing workload might lead to undesired results such as lower adenoma detection rates per colonoscopy [11] and longer waiting times for the colonoscopy [12]. As shown in [12] the median weighting time for the screening colonoscopy is 210 days with the maximum waiting time equaling 631 days, in Canada. Long waiting time not only increases the patient’s mental burden and the risk of precancerous polyps’ evolvement.”
Reference:
Greenspan, M., Prickett, E., & Melson, J.: High Clinical Patient Workload Leads to Increased Premature Adenomatous Polyp Surveillance Colonoscopy: 1391. Official journal of the American College of Gastroenterology| ACG, 2015, 110, S601.
Almadi, M. A., Sewitch, M., Barkun, A. N., Martel, M., & Joseph, L.: Adenoma detection rates decline with increasing procedural hours in an endoscopist’s workload. Canadian Journal of Gastroenterology and Hepatology, 2015, 29(6), 304-308.
Sey, M. S. L., Gregor, J., Adams, P., Khanna, N., Vinden, C., Driman, D., & Chande, N. (2012). Wait times for diagnostic colonoscopy among outpatients with colorectal cancer: a comparison with
Canadian Association of Gastroenterology targets. Canadian Journal of Gastroenterology, 26(12), 894-896.12
- line 49: 'healthcare centres NEED TO identify patients who require colonoscopy using noninvasive examination' sounds too strong and there is no evidence for this until an appropriate tool is discovered and implemented. so please make a softer statement (i.e. it would be ideal, practical, etc).
A: Authors thank the reviewer for the comment. We do agree that sentence is a strong statement. Therefore, we revised the sentence. It is now “T Therefore, healthcare centers are actively searching for a risk stratification tool that identifies patients who require colonoscopy using noninvasive examination results.”
- Authors don't mention if the included patients had previous polypectomies or not. is this an exclusion criteria? have you taken into account this when building the tool?
A: Authors thank the reviewer for the comment. Since this is an annual healthcare check-up data, the “previous polypectomies or not” is not the item in the check-up data set. We don’t have access to such data and the doctors in the annual check-up section also cannot obtain such data in real-time. Since the purpose of this study is to develop a risk stratification tool that can be applied on-site at the annual check-up outpatient unit, we do not use any data that cannot be obtained in real-time at the annual check-up outpatient unit.
- METHODS
- seems that authors have excluded patients with BMI>35. please explain why
A: Since our data of colonoscopy diagnosis is conducted under anaesthetization. The anesthesiologist uses the BMI>35 as exclusion criteria, hence, we don’t have any patient whose BMI is greater than 35. We have added this in the future study as one of the possible extensions for the future study. Specifically, we wrote “3) our sigmoidoscopy is conducted under anaesthetization. Thus, our data set excluded patients with BMI>35 due to the protocol code of the anesthesiologist. Future researchers can build an RF model for this sub-group or collect data of non-anesthetized sigmoidoscopy diagnostics”
- It would be nice to have a list of all those risk factors taken into account for the analysis, before the identified ones.
A: Authors thank the reviewer for the comment. We added a new section , section 2.3 to address this comment. In the new section 2.3 “risk stratification tool building” we wrote “As described in section 2.1, all items in the annual check-up data are collected for this research. Based on the previous research [13-17] we selected risk factors from the following categories as 1) patient’s demographic characteristics including age, sex, weight, and height. 2) Patient’s medical history including hypertension, diabetes, and Helicobacter pylori infection 3) The colonoscopy diagnosis results including colorectal polyps, ulcerative colitis, hemorrhoids; intestinal hemorrhage, etc. 4) Abdominal ultrasonography diagnosis including GB polyps, GB stones. 5) Blood sample diagnosis results including fasting blood glucose total cholesterol, high and low-density lipoprotein (HDL and LDL), triglycerides, etc.. 5) Dental diagnosis results including periodontitis; periodontal disease; chronic periodontitis, and chronic gingivitis. All of the diagnosis results are binary the data with the 1= positive diagnosed, 0 = otherwise. BMI is calculated based on the weight of height of the patient. Furthermore, patients’ demographic data are dichotomized into binary or categorical data. Age is dichotomized as over/under 50 years old greater or less than 50 years old, and BMI is categorized as (underweight [<18.5 kg/m2], normal [18.5–25 kg/m2], overweight [25–30 kg/m2], and obese [>30 kg/m2].” We also provide full risk selection procedure as in “The Heuristic” in the new section 2.3. Please refer to the manuscript for more details.
- Results
- the results of the 2 forest plot have not been explained within the results
A: Authors thanks reviewer for the comment. We have restructured the presentation of the paper and moved the explanation of the forest charts in the result section. We wrote “Figures 2 and 3 further demonstrate the significance and positive or negative impacts of each risk factor, respectively. To construct a risk stratification tool based on these risk factors, a random forest machine learning method was employed. In our study, age, Helicobacter pylori infection, and hypertension were all risk factors for colorectal polyps. A forest chart was also constructed to present the estimated odds ratios for each risk factor, as shown in Figures 1 2 and 23. While traditional statistical methods such as logistic regression have an AUC >0.5, the discriminability is not as high as healthcare centers may wish (0.5086–0.5900). Therefore, a machine learning method is required to build a model with higher discriminability. As shown in Figures 1 2 and 23, abnormal body mass, age, and Helicobacter pylori are the most influential risk factors for colorectal polyps, regardless of the patient’s gender. We also found that hypertension was a significant risk factor for colorectal polyps in male patients. Moreover, the influence of different abnormal body masses was significantly different between gender and age groups. Thus, we further divided the patients according to age, gender, and body mass to obtain 16 patient subgroups (2 × 2 × 4) (as shown in Table 3). Since risk factors differ according to age and sex, a risk stratification model was built for each group of patients. For each subgroup, a risk stratification tool was built via a machine learning method. Building a patient-characteristic-specific risk stratification model through the machine learning method not only enhances the discriminability of the model but also identifies a set of more precise risk factors for each patient group. Healthcare centers can utilize these risk factors to precisely diagnose patients with colorectal polyps”
- Table 2 should be better explained
A: Authors thanks reviewer for the comment. We added a paragraph to explain the result summarized in table 2. More specifically we wrote “In Table 2, we find that the risk factors differ based on gender, age, and BMI. Therefore, all patients were divided into sub-groups based on gender, age, and BMI. For each group, risk factors for GB polyps, hypertension, tooth disease, and Helicobacter pylori infection were input as independent variables to predict colorectal polyps. In Table 2 we presented the AUC of risk factors with p-values of the model and AUC from the logistics equations, where the p-values are less than 0.1 for at least male or female. Results of total cholesterol, high lipoprotein cholesterol, and triglycerides are excluded since their p-values are greater than 0.1. As we can observe from Table 2, the observed significances (p-values) for risk factors are different from male to female. Thus, we separate the patients with their gender for the machine learning stage. While in table 2 we did not examine the p-value for different BMI levels, previous literature suggests BMI might significantly relate to the evolvement of colorectal polyps. For example, [27] found that the overweight and underweight statuses are significantly correlated with the gut microbiota and metabolism. Jain et al [28] found that obesity significantly im-pacts metabolism and is accessible with colorectal cancer and polyps. Hence, we also separate patients with their status of BMI.”
Reference
Wan, Y., Yuan, J., Li, J., Li, H., Yin, K., Wang, F., & Li, D.: Overweight and underweight status are linked to specific gut microbiota and intestinal tricarboxylic acid cycle intermediates. Clinical Nutrition, 2020,39(10), 3189-3198.
Jain, R., Pickens, C. A., & Fenton, J. I. The role of the lipidome in obesity-mediated colon cancer risk. The Journal of nutritional biochemistry, 2018, 59, 1-9.
- DISCUSSION
- the discrepancies in results regarding BMI needs further explanation. why the authors think it has such an effect on the AUC? please discuss this factor
A: Authors thanks reviwer to bring up this point. We address this comment in two aspects. First, in the result section we further explain the reasoning of using BMIT to group patients. We wrote: “…While in table 2 we did not examine the p-value for different BMI levels, previous literature suggests BMI might significantly relate to the evolvement of colorectal polyps. For example, [27] found that the overweight and underweight statuses are significantly correlated with the gut microbiota and metabolism. Jain et al [28] found that obesity significantly impacts metabolism and is accessible with colorectal cancer and polyps. Hence, we also separate patients with their status of BMI.”
Secondly, we further discuss the impact of grouping patients with their BMI status. In this discussion, we wrote “As discussed in [27] the BMI status, both over- and underweight, can alter the gut metabolism, and as [28] point out the change in metabolism significantly relate to colorectal cancer and polyps. We hypothesis that the BMI has a significant indicator of different colorectal health therefore, the risk factor might change from one BMI status to another. The results of AUC prove that our hypothesis is correct. For some BMI status, it is easier to identify the patient with colorectal polyps, and others are not. The risk factors also differ from one BMI status to another.
- a comparison between the authors' tool and previous ones developed deserved further discussion. what are the differences? what are the advantages of the authors' proposal?
A: Authors thanks reviewer for bringing up this point. Our work is the first to develop a risk stratification tool for colorectal polyps. The previous study in the field of ML/AI is liver-related non-invasive risk stratification tools. Goldman et al (2020) and Fialoke et al (2018) both use annual health check-up data to classify healthy patients and patients with liver disease (NAFLD; NASH; cirrhosis). Their AUC is in the range of 0.8486 to 0.8764. While this is not a truly compatible comparison, we believe our result is not worse than any other risk stratification tools using annual check-up data.
We also address this comment in the manuscript.
We wrote “In our study, there is no doubt that all aforementioned risk factors are noninvasive indicators of colorectal polyp formation [48]. Our risk stratification tool, which is built based on identified risk factors with a machine learning method, exhibits high sensitivities (70%–80%) compared with that in noninvasive tools developed by previous studies (60%–70%) [55]. Other decision tree-based studies [25, 26] which build noninvasive stratification tools using annual check-up data for non-alcoholic fatty liver obtained AUC ranges from 85%-87%. Compare with previous studies, the proposed model outperformed in several subgroups, such as elder obsess individuals”
Reference
Park W, Lee H, Kim EH, Yoon JY, Park JC, Shin SK, Lee SK, Lee YC, Kim WH, Noh SH: Metabolic syndrome is an independent risk factor for synchronous colorectal neoplasm in patients with gastric neoplasm. Journal of gastroenterology and hepatology 2012, 27(9):1490-1497.
Tanwar, S., & Vijayalakshmi, S. Comparative analysis and proposal of deep learning based colorectal cancer polyps classification technique. Journal of Computational and Theoretical, Nanoscience 2020, 17.5: 2354-2362.
Goldman O, Ben-Assuli O, Rogowski O, Zeltser D, Shapira I, Berliner S, Zelber-Sagi S, Shenhar-Tsarfaty S. Non-alcoholic Fatty Liver and Liver Fibrosis Predictive Analytics: Risk Prediction and Machine Learning Techniques for Improved Preventive Medicine. J Med Syst 2021; 45: 22.
Fialoke S, Malarstig A, Miller MR, Dumitriu A. Application of Machine Learning Methods to Predict Non-Alcoholic Steatohepatitis (NASH) in Non-Alcoholic Fatty Liver (NAFL) Patients. AMIA Annu Symp Proc 2018; 2018: 430-439.
- Conclusion
- the authors don't mention the difference between male/female within the conclusion. do they recommend the colonoscopy verification in the same cases?
A: Authors thanks reviewer for the comment. We address this comment by adding the following paragraph in the conclusion to discuss our suggestion regarding gender. “For obese females, GB polyps warrant further colonoscopy verification. For males over age 50 and not underweight, hypertension is a strong indicator of possible colorectal polyps. We also find that for either underweight or obese patients, the AUC is higher than other groups” and “This is verified that our design of grouping patients with different demographic characteristics before building a machine learning model. ”
- And what about the BMI? does it need to be taken into account? please clarify
A: Authors thanks reviewer for the comment. To address this comment, we revised our conclusion section with the following changes:
We wrote “In this research, we propose a new approach of building risk stratification tool for colorectal polyps. First, we identified a set of promising risk factors using traditional statistical analysis such as z-test and logistics regression. We find that the risk factors significantly differ for different genders, ages, and BMI statuses. Then we separate patients with key demographic characteristics which we believe each subgroup has a different set of risk factors. Then we implement the random forest to build the machine learning model to stratify patients with and without colorectal polyps.”
And
“We also find that for either underweight or obese patients, the AUC is higher than other groups. That is abnormal weight is a strong indicator of health status, and different health statuses should be modeled differently. This is verified that our design of grouping patients with different demographic characteristics before building a machine learning model.”
Again authors thanks reviewer for your time and effort in reviewing this work!

Reviewer 2 Report
Authors have conducted an interesting study, but its overall presentation is poor and need to be very well revised.
Authors should discuss and compare state of the art, as it is important to know about the similar work done, and how author's achievements are significant.
In methods section, author have discussed application of Random Forest but without valid rational. Furthermore, its important to know what alternative options could be. Furthermore, variable assignments to the algorithm are not well addressed, and training model details are not well justified.
Authors should guide readers to the source code availability to test and reproduce their results.
Authors have denied sharing of data and given reason that it is due to the protection of patient’s privacy, which isn’t the case because data has been used in de-identified fashion and it must be available to community along with their source code, if they would publish it.
Overall writing is poor, presentation of results need to be improved as well as data visualization must be adapted. Furthermore, citation can also be improved.
Author Response
Response Letter
Reviewer 2
2021/12/16
Dear Reviewer,
The authors would like to thank the reviewer for your effort in reviewing this manuscript. Your constructive comments help us to improve the quality of this paper. We would also like to wish the reviewer a happy Holiday Season! The reviewer’s comments are boldfaced, and our answer is in blue.
Authors have conducted an interesting study, but its overall presentation is poor and need to be very well revised.
A: Authors thanks reviewer for the comment. We did the following major revision to address this comment along with all other comments.
In section 1
We add a paragraph to elaborate on why the risk stratification tool for the colorectal polyp is needed.
In section 2
- We added a new section 2.3 to describe our overall modeling and risk stratification tool building procedure.
- To clean section 2, we remove the material related to empirical results from section 3.
- We rewrote most sections 2.5 and parts of 2.4 to present details in the risk factor selection and modeling building process.
- We provide two pseudo codes including one model building heuristic and one machine learning algorithm to better illustrate our methods.
In section 3
- We restructure our results so that the presentation is more fluent.
- More details are added to elaborate on our results present in the results and our risk stratification tool.
In section 4
- We elaborate on why grouping patients with their BMI can improve the AUC of the risk stratification tool.
- We add compassion between our model and previous studies.
Authors should discuss and compare state of the art, as it is important to know about the similar work done, and how author's achievements are significant.
A: Authors thanks reviewer for bringing up this point. Our work is the first to develop a risk stratification tool for colorectal polyps. The previous study in the field of ML/AI is liver-related non-invasive risk stratification tools. Goldman et al (2020) and Fialoke et al (2018) both use annual health check-up data to classify healthy patients and patients with liver disease (NAFLD; NASH; cirrhosis). Their AUC is in the range of 0.8486 to 0.8764. While this is not a truly compatible comparison, we believe our result is not worse than any other risk stratification tools using annual check-up data.
We also address this comment in the manuscript.
We wrote “In our study, there is no doubt that all aforementioned risk factors are noninvasive indicators of colorectal polyp formation [48]. Our risk stratification tool, which is built based on identified risk factors with a machine learning method, exhibits high sensitivities (70%–80%) compared with that in noninvasive tools developed by previous studies (60%–70%) [55]. Other decision tree-based studies [25, 26] which build noninvasive stratification tools using annual check-up data for non-alcoholic fatty liver obtained AUC ranges from 85%-87%. Compare with previous studies, the proposed model outperformed in several subgroups, such as elder obsess individuals”
Reference
Park W, Lee H, Kim EH, Yoon JY, Park JC, Shin SK, Lee SK, Lee YC, Kim WH, Noh SH: Metabolic syndrome is an independent risk factor for synchronous colorectal neoplasm in patients with gastric neoplasm. Journal of gastroenterology and hepatology 2012, 27(9):1490-1497.
Tanwar, S., & Vijayalakshmi, S. Comparative analysis and proposal of deep learning based colorectal cancer polyps classification technique. Journal of Computational and Theoretical, Nanoscience 2020, 17.5: 2354-2362.
Goldman O, Ben-Assuli O, Rogowski O, Zeltser D, Shapira I, Berliner S, Zelber-Sagi S, Shenhar-Tsarfaty S. Non-alcoholic Fatty Liver and Liver Fibrosis Predictive Analytics: Risk Prediction and Machine Learning Techniques for Improved Preventive Medicine. J Med Syst 2021; 45: 22.
Fialoke S, Malarstig A, Miller MR, Dumitriu A. Application of Machine Learning Methods to Predict Non-Alcoholic Steatohepatitis (NASH) in Non-Alcoholic Fatty Liver (NAFL) Patients. AMIA Annu Symp Proc 2018; 2018: 430-439.
In methods section, author have discussed application of Random Forest but without valid rational. Furthermore, its important to know what alternative options could be. Furthermore, variable assignments to the algorithm are not well addressed, and training model details are not well justified.
Authors should guide readers to the source code availability to test and reproduce their results.
A: Authors thank the reviewer for this comment. We added a new section 2.3 to introduce our modeling procedure and section 2.5 provides pseudo-code for our machine learning algorithm. We detailed all information used in our source code to adjust all details in the modeling and data training procedure in the pseudo-codes. We believe that pseudo-code is a better and cleaner way to present our overall procedure. All readers can use our pseudo-code as a guide to recreate our model. More specifically we added”
The Heuristic:
Step 1: Collect data from annual health check-ups. All risk factors are indexed from i=1…N, the value of the risk factor is xi , where there are N risk factors in total.
Step 2 Pre-screen with a z-test for two sample proportions with a significance level equal to 0.05 is applied to select potential risk factors. Where the two sample proportions are calculated as
For all risk factor i,
phi = the proportion of patients who has colorectal polyps for patients with risk factor xi=h-1.
That is,
p1i = the proportion of patients who has colorectal polyps for patients with risk factor xi=0.
p2i = the proportion of patients who has colorectal polyps for patients with risk factor xi=1.
Step 3 The null and alternative hypothesis is stated as below:
Null Hypothesis: p1i = p2i=…. phi
We record all risk factors which has a significantly different sample proportion between patients with and without colorectal polyps.
Step 4 Logistics regression is applied for each risk factor to calculate the discriminability for each risk factor. Based on the logistic regression, we identified the demographic risk factors which can segregate patients into different sub-groups for the machine learning process.
Step 4 Machine learning is applied to each sub-group to construct the risk stratification tool.
Step 5 We output the system of models which consisted of multiple random forest models.
Step 6 Output our four-fold-cross validation.
And
Machine Learning Algorithm (RF):
Step 1: Input all risk factors as vector X=< x1……xh> and the y= 1 if a patient is diagnosed with colorectal polyps, and zero otherwise. Also, input the demographic factors for aggregating patients into subgroups. Go to step 2.
Step 2: Segregate all patients into subgroups. Index subgroups as k = 1…N for N groups in total. Let k =1 and go to step 3.
Step 3: Input all risk factors X and y in the kth sub-group. Go to step 4.
Step 4: Input all data in with path_name = group k, with the following specification of random forest package in python. We selected the four-fold validation, thus 75% of data will be randomly selected for modeling building and 25% will be reserved for validation. For each run, the random forest will repeat four times for validation. Output the model and go to step 5.
Branch criterion: gini index
Number of estimators: 1000
Min_samples_leaf=5
Class weight: balanced
Validation: Four-fold
Calculate the following statistics:
Specificity = True negative /(true negative + false positive)
Sensitivity = True positive /(true positive + false negative)
Area Under Curve (AUC)
Step 5: Collected the outputted model and check if k = N, if not let k =k+1 and go to step 3, otherwise end the algorithm.
Authors have denied sharing of data and given reason that it is due to the protection of patient’s privacy, which isn’t the case because data has been used in de-identified fashion and it must be available to community along with their source code, if they would publish it.
To address the question about data visibility, unfortunately, we cannot make the data available to the public. The data used in this research is approved by the Ethics Committee of the Institutional Review Board of Chang Gung Memorial Hospital. While the committee approves our research team to analyze the data and publish our results. However, the Ethics Committee doesn’t approve any other usage of data including making it available for the public. Therefore, we cannot publish the data or make it available to the public. However, we do include a snapshot of part of our data with this response letter, and we can make it as an appendix for the manuscript if needed for data transparency.
Overall writing is poor, presentation of results need to be improved as well as data visualization must be adapted. Furthermore, citation can also be improved.
A: Authors thanks reviewer for the comment. We address this comment by thoroughly proofread this manuscript with help from professional writers. We also restructured and add more new material in this revision as discussed in the response to the first comment.
In the result section,
- we move the old figure 1 and 2 to the result section and add a paragraph to discuss it.
- we add more discussion for table 2 in the result section.
More specifically, we added
“n Table 2, we find that the risk factors differ based on gender, age, and BMI. Therefore, all patients were divided into sub-groups based on gender, age, and BMI. For each group, risk factors for GB polyps, hypertension, tooth disease, and Helicobacter pylori infection were input as independent variables to predict colorectal polyps. In Table 2 we presented the AUC of risk factors with p-values of the model and AUC from the logistics equations, where the p-values are less than 0.1 for at least male or female. Results of total cholesterol, high lipoprotein cholesterol, and triglycerides are excluded since their p-values are greater than 0.1. As we can observe from Table 2, the observed significances (p-values) for risk factors are different from male to female. Thus, we separate the patients with their gender for the machine learning stage. While in table 2 we did not examine the p-value for different BMI levels, previous literature suggests BMI might significantly relate to the evolvement of colorectal polyps. For example, [27] found that the overweight and underweight statuses are significantly correlated with the gut microbiota and metabolism. Jain et al [28] found that obesity significantly im-pacts metabolism and is accessible with colorectal cancer and polyps. Hence, we also separate patients with their status of BMI.”
“Based on our results in section 3.1, we separate all patients into 16 groups via their age, gender, and BMI status. The random forest algorithm in section 2.5 is applied to each group and the validation results are summarized in Table 3. The input risk factors include hypertension, chronic periodontitis, humanoids, Helicobacter pylori infection, GB stones and polyps, and diabetes. However not all risk factors are significant in the final model, and the performance of the stratification model varied extensively.”
We also add the following citations to enrich this study.
Greenspan, M., Prickett, E., & Melson, J.: High Clinical Patient Workload Leads to Increased Premature Adenomatous Polyp Surveillance Colonoscopy: 1391. Official journal of the American College of Gastroenterology| ACG, 2015, 110, S601.
Almadi, M. A., Sewitch, M., Barkun, A. N., Martel, M., & Joseph, L.: Adenoma detection rates decline with increasing procedural hours in an endoscopist’s workload. Canadian Journal of Gastroenterology and Hepatology, 2015, 29(6), 304-308.
Sey, M. S. L., Gregor, J., Adams, P., Khanna, N., Vinden, C., Driman, D., & Chande, N. (2012). Wait times for diagnostic colonoscopy among outpatients with colorectal cancer: a comparison with Canadian Association of Gastroenterology targets. Canadian Journal of Gastroenterology, 26(12), 894-896.12
Abdulqader, D. M., Abdulazeez, A. M., & Zeebaree, D. Q.: Machine learning supervised algorithms of gene selection: A review. Machine Learning, 2020, 62(03).
Goldman O, Ben-Assuli O, Rogowski O, Zeltser D, Shapira I, Berliner S, Zelber-Sagi S, Shenhar-Tsarfaty S. Non-alcoholic Fatty Liver and Liver Fibrosis Predictive Analytics: Risk Prediction and Machine Learning Techniques for Improved Preventive Medicine. J Med Syst 2021; 45: 22.
Fialoke S, Malarstig A, Miller MR, Dumitriu A. Application of Machine Learning Methods to Predict Non-Alcoholic Steatohepatitis (NASH) in Non-Alcoholic Fatty Liver (NAFL) Patients. AMIA Annu Symp Proc 2018; 2018: 430-439.
Wan, Y., Yuan, J., Li, J., Li, H., Yin, K., Wang, F., & Li, D.: Overweight and underweight status are linked to specific gut microbiota and intestinal tricarboxylic acid cycle intermediates. Clinical Nutrition, 2020,39(10), 3189-3198.
Jain, R., Pickens, C. A., & Fenton, J. I. The role of the lipidome in obesity-mediated colon cancer risk. The Journal of nutritional biochemistry, 2018, 59, 1-9.
Again, the authors thanks to the reviewer for your effort in reviewing this manuscript!
